# Synthesized Approach for Evaluating the Integral Suspension Pressure (ISP) Method and the Hydrometer in the Determination of Particle Size Distribution

Maria Camila Olarte [1], Juan Carlos Ruge [1,*] and Juan Félix Rodriguez-Rebolledo [2]

1   Programa de Ingeniería Civil, Facultad de Ingeniería, Universidad Católica de Colombia,
    Bogotá 111311, Colombia
2   Programa de Pós-Graduação em Geotecnia, Universidade de Brasília, Brasilia 70910-900, Brazil
*   Correspondence: jcruge@ucatolica.edu.co

**Abstract:** Different techniques have been developed in the 20th and 21st centuries to address the study of particle size distribution in fine materials. Most of these techniques are based on gravitational sedimentation processes. The modern method used in this research bases the measurement on the change in pressure of the aqueous medium caused by the progressive settling of fine particles. Different materials were evaluated within the study to compare the results of the integral suspension pressure (ISP) method with a traditional approach used worldwide, such as hydrometers. Although the ISP method is considered promising and reliable in measuring the particle size distribution of the fine fraction, current literature lacks comparisons with traditional methods. This aspect would help in the definitive validation of the technique and its use in practical engineering. The hydrometer recorded silt content was greater than ISP over the whole range of measurements, especially in yellow kaolin, in which the silt content is more than 40%. Compared to the hydrometer method, the ISP presented a tendency to misclassify the soil texture of bentonite due to the high clay content. The considerable differences, especially in clays with finer particles such as bentonite, demonstrate that the modern ISP technique can detect very fine projected materials within the particle size distribution compared to conventional methodologies. The study's objective is mainly to compare both methods, given the important technological evolution that the ISP method presents in relation to the hydrometer.

**Keywords:** integral suspension pressure method; hydrometer; clay

## 1. Introduction

Research at the textural level of particles is one of the most common physical analyses performed on soil samples, and this property significantly influences the other intrinsic constituents of sediments, such as chemical-mechanical properties [1,2], hydrophysical characteristics, and pedotransfer functions concerning hydraulic properties [3,4]. The particle size distribution (PSD) determines the separation percentages of clay, silt, and sand through mechanical fractionation by direct separation in grains with diameters greater than 75 μm (according to ASTM) and by progressive deposition processes for configurations smaller than 75 μm. Gravitational settling analysis is adequately described by Stokes' law [5], where the hydraulic principle of microparticle movement in water and variability in suspension density are considered [6–9]. On this basis, an equivalent diameter is assumed for each particle, similar to the diameter of a sphere of equal density, whose settling velocity is proportional to the vertical velocity in the suspension. The terminal fall velocity of each particle is given by the effective fall distance and the time interval allotted for the trajectory.

Typically, standard methods for grain size analysis are established from several physical principles. Methodologies such as laser diffraction (DLM), where the grain size is a function of the cross-sectional area of the particle in a two-dimensional plane, and PSD

determination by image analysis have the advantage of covering a wide range of grains, although their principles are primarily optical [10]. Techniques supported on the physical principle of impedance include particle counting by electroresistance, e.g., Coulter counter [11], and transition time. Versions based on the sedimentation principle are, for example, the Atterberg method, centrifugal sedimentation [12], sedigraph, hydrophotometer, pipette, and hydrometer (Figure 1). In [13], the authors summarize the most commonly used experimental methods for the granulometric analysis of soil.

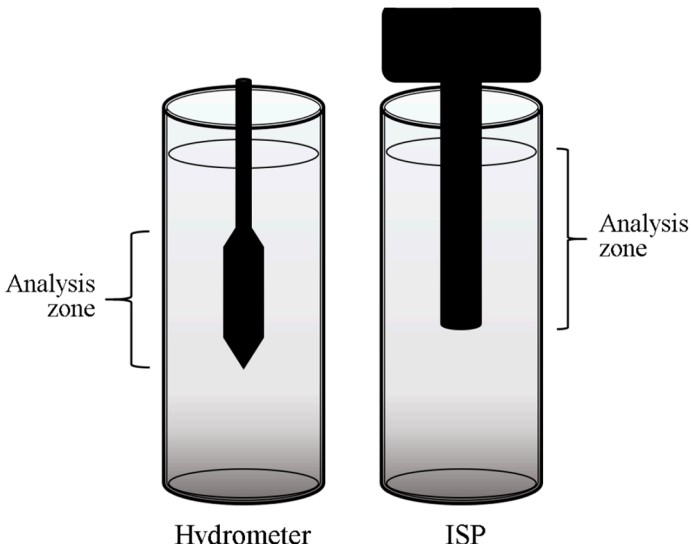

**Figure 1.** Representation of the hydrometer and the ISP methods.

The determination of PSD by induced sedimentation processes has been carried out for almost a century [14,15]. The initial versions estimated the intrinsic PSD frame through the weight percentage of soil in a known volume of water or through the concentration of soil particles in suspension [3,16]. Traditionally, pipettes are the most accepted and accurate method for PSD construction [17]. Based on the preset sampling depths, the mechanism allows for the assignment of specific fraction sizes to predetermined extraction times. However, both the execution process and data curation require a considerable amount of precise work [18,19].

The hydrometer procedure as a means of PSD quantification was introduced in the literature in 1927 by Bouyoucos [20–22]. In contrast to the pipette, the hydrometer floats to a depth that depends exclusively on the concentration of the suspension [23]. The procedural methodology is comparatively simple since absolute settling of all sand and silt particles beyond the settling depth is required for PSD assembly. In other words, suspension decreases with a terminal settling velocity proportional to the square of its equivalent diameter. Depending on the intrinsic PSD of the soil sample, the suspension density decreases continuously with time as larger particles, such as sands and silts, exceed the measurement depth. However, because the hydrometer's own effective fall distance changes during the settlement, the percentage size distribution obtained may not be as expected if measurements are made at a predetermined depth [24,25].

Generally, modern adaptations for defining PSD are characterized by associating high repeatability of results with high degrees of simplicity in execution. The extended integral suspension pressure (ISP) method [26,27] focuses on derivative fundamentals to calculate PSD as a function of the temporal change in pressure within the suspension. The PSD of a sample is determined by adjusting the simulated and observed pressure time series using global optimization mathematical models. Similar to the hydrometer and pipette, the ISP is based on the sedimentation principle, although it does not directly disturb the gravitational settling process of the particles (Figure 1). Although the diametric spectrum covered by the

ISP is significantly larger than that covered by the hydrometer or the pipette, little research has considered its implementation [28–31].

Techniques based on the principle of sedimentation, such as the hydrometer or ISP, define PSD in different ways, and the analytical device used to determine the size and associated particle populations can produce varying behavior from identical specimens [32]. Additionally, since many of these methods measure different properties of the same type of material, the operating system can produce variable data from identical samples. Therefore, during recent decades, numerous comparative studies between PSD measurement techniques have been documented. In [33], the authors performed an extensive comparative analysis of four physical measurement principles (laser diffraction, sedimentation, impedance, and optical) across ten instruments, traditional and contemporary, to calculate the diametric frame of PSD in clayey sediments. A summary of comparative approaches to determine PSD is provided in Table 1.

**Table 1.** Summary of comparative approaches to determine PSD using ISP.

| Ref. | Methods | Physical Principles Evaluated |
|------|---------|-------------------------------|
| [34] | ISP and hydrometer | Sedimentation |
| [26] | ISP and pipette | Sedimentation |
| [35] | ISP and sedimentation | Sedimentation |

Today the existing studies that attempt to establish a relationship between hydrometer and ISP are meager. Practically, the only studies that involve ISP and hydrometer in terms of direct comparison are those articulated in references [34,35]. Although it is a relatively recent methodology, the studies carried out show promising results. However, compared with a traditional technique, such as hydrometry, it reveals some discrepancies in the data, especially in clay-type fractions. The ISP has an advantage over the hydrometer, which is its high sensitivity to detect tiny particles due to its sensor and allows a pseudocontinuous curve. The hydrometer, as previously mentioned, presents deficiencies in very plastic clays, with the presence of montmorillonite. It is still necessary to carry out more investigations that allow comparing, with high reliability, the ISP with classical methods such as hydrometer and pipette.

This particular research analyzes a possible correlation between both methods for the materials used. Although the two methods are based on the classical sedimentation theory, the data capture is very different and sensitive when obtaining PSD results. However, an ambitious study of samples and materials is required to reach these types of conclusions. The analysis was carried out using three different soil typologies, all with different textures. Thus, the subsequent information represents the first comparative analysis between ISP and hydrometer measurements performed on soil samples of different origins.

## 2. Materials and Methods

### 2.1. Physical Properties of the Soil

Three soil typologies were collected from both natural and anthropogenic environments across a wide range of climatological, geomorphological, and pedogenetic settings, as shown in Figure 2. The soils were collected from three areas: (i) kaolin mine, (ii) Candelaria desert, and (iii) industry, all near the city of Tunja, in the department of Boyacá, Colombia. The soils tested were carried out on three inorganic soils, including bentonite, and white and yellow kaolin ($k_w$, $k_y$, respectively). These soils were chosen to cover a different textural range. Based on ASTM D854-14 [36] for the calculation of the specific densities of the sample solids through a water pycnometer, the results of specific gravity (Gs) were required for the sedimentation analyses. The results are shown in Table 2.

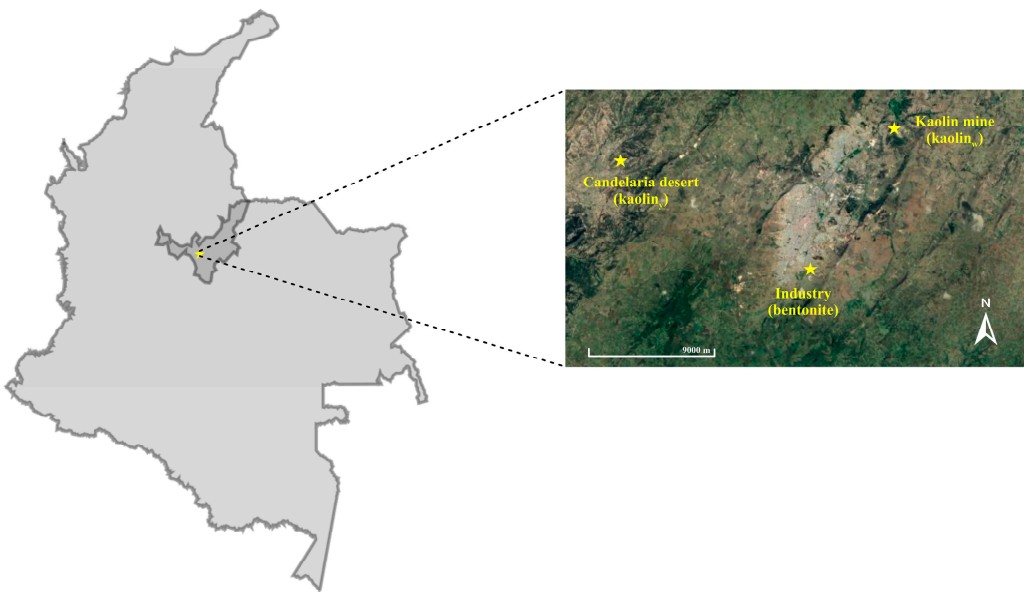

**Figure 2.** Location of the study area in Colombia and sampling sites.

**Table 2.** Gs of the soil samples.

| Soil | Gs |
|---|---|
| Bentonite | 2.70 |
| Kaolin$_y$ | 2.54 |
| Kaolin$_w$ | 2.58 |

The changes in the textural information of each type of soil studied within the discrete and mixed multiphase layers were focused on using the basal reflections obtained employing X-ray diffraction (XRD). These results are presented in Table 3. Additionally, the content of chemical macro- and microelements present in the polymeric matrix of the clay core was explored using X-ray fluorescence (XRF) spectra. This information is summarized in Table 4.

**Table 3.** Mineralogical composition by XRD.

| Clay Minerals/Sample | Kaolin$_w$ | Bentonite | Kaolin$_y$ |
|---|---|---|---|
| Kaolinite | 89.17 | 6.23 | 75.35 |
| Illite | 9.75 | —— | 16.00 |
| Montmorillonite | 1.08 | 93.43 | 6.74 |

**Table 4.** Composes (% $p/p$) according to the results of XRF.

| Sample | SiO$_2$ | Al$_2$O$_3$ | Fe$_2$O$_3$ | CaO | K$_2$O | MgO | Na$_2$O | P$_2$O$_5$ | TiO$_2$ |
|---|---|---|---|---|---|---|---|---|---|
| Kaolin$_w$ | 61.16 | 18.92 | 3.71 | 0.12 | 1.70 | 1.67 | 0.91 | 0.95 | 0.53 |
| Bentonite | 48.08 | 19.08 | 6.37 | 1.26 | 0.76 | 3.04 | 2.62 | 0.09 | 0.88 |
| Kaolin$_y$ | 59.23 | 17.34 | 4.65 | 1.81 | 0.29 | 3.72 | 1.45 | 0.13 | 0.36 |

To provide diameters less than 75 μm [37], all samples from each soil group were passed through a No. 200 sieve for both hydrometry and ISP methods. All soils were tested using the hydrometer and ISP techniques described below. The experiments were carried out in distilled water and were interpreted by three independent experimental sets of tests per soil. Thus, 20 measurements were performed with each technique, i.e., 40 grain size analyses in total. The separation criteria for clay (<2 μm), silt (2–75 μm), as well as the measurement techniques, were performed following the ASTM D7928-21 standard [38].

### 2.2. Hydrometer Method

The hydrometer used for the dispersed particle concentration controls was of the ASTM-H152-161 68°F type. This is a standard model and is internationally approved. A suspension smaller than 75 μm was dispersed with sodium hexametaphosphate (SHMP) at a rate of 50 g per liter of solution, according to the user manual. To measure the particular concentration in suspension, the ASTM procedure requires the use of a compound correction for each hydrometer reading, necessary both to neglect the increase in fluid density ($\rho_i$) due to the addition of SHMP and to compensate for readings taken at the top of the meniscus rather than the bottom. Because the settling probe with the 5% SHMP solution is placed in constant temperature reservoirs, a linear composite correction–temperature relationship is assumed to exist between the two measured points [38]. The measurements were recorded at 1, 2, 5, 15, 30, 60, 120, 250 and 1440 min. Using the compound correction derived from the measurements, the solution temperature, the specific gravity, and the actual hydrometer reading, the percentage of soil in suspension is determined by the (Equation (1)) [39]:

$$P = (R_{ASTM}{}^a / Ms) \tag{1}$$

where $R_{ASTM}$ [gL$^{-1}$] is the corrected reading, a is a nondimensional correction factor for the difference between densities, and $Ms$ [gL$^{-1}$] is the dry weight of the soil sample per liter of suspension. For the calculation of the shape and dimension of the suspended bodies, the diameter $d$ (mm) is a function of both the terminal fall time of each particle and the effective depth of the hydrometer as follows (Equation (2)):

$$d = K\sqrt{(L/t)} \tag{2}$$

$K$ is a constant dependent on the density of the suspended particles and on the viscosity of the fluid. A standardized table of $K$ values is available in Table 3 of ASTM D422-63 [40], $t$ is the precipitation time in minutes and $L$ (cm) is the effective measuring depth of the hydrometer prescribed in terms of the uncorrected reading.

### 2.3. Integral Suspension Pressure (ISP) Method

To evaluate the ISP measurement method, the particle size distribution (PSD) was estimated for each of the three replicate tests per soil. It is understood that the pretest phase is key to successful results in the ISP. It is crucial to allow an adequate collection of Gs between these steps. This parameter is essential since it is an input parameter in the software of the technique addressed in this study. The Gs was found with the oven-dried sample to guarantee the total loss of water, and the mass of the mineral was exact in the calculation. Some of the pretreatment stages are optional, such as removing organic matter due to the possible oxidation that causes bonding between the particles when hydrogen peroxide is used. In this study, the deflocculant used was sodium hexametaphosphate, which does not show this problem and is more effective for dispersing clay particles because it incorporates sodium ions.

Due to the insignificant presence of iron oxides and soluble salts in the samples, which can be evidenced in the XRD and XRF tests in Tables 3 and 4, respectively, the stages of removal of soluble salts, plaster, and iron oxides recommended by the supplier were not carried out. The samples were wet sieved through a 75 μm sieve using the ASTM D1140-17 [37] to guarantee the total presence of fines in the sample and to eliminate any component of sand that could eventually hinder the analysis. Fine particles (<75 μm) collected after sieving were transferred to standard cylinders for ISP analysis and then placed in a 22 °C constant temperature water bath. However, the executable precepts of the ISP are established by the DIN 18,123 [41]. In each apparatus, before the first reading, a one-minute homogenization process is carried out employing manual agitation that consists of rocking the assembly. This is to remove sedimented residues from the bottom of the container and to make the suspended sample more uniform. In some cases, using a stirrer inserted into the container is valid, and the specimen can be appropriately mixed.

The time spent between mixing and insertion of the sensor should not exceed 20 s since the pressure measurement is done every 10 s in the internal system of the apparatus, but the first value used in the time series of 30 s.

The methodology of which the intrinsic mechanism of the ISP is part does not require additional adjustments, as in conventional techniques. The device comprises a pressure sensor that is connected to a measuring head and added to a temperature meter. Both readings are recorded in the head of the device and sent to the data acquisition via USB (Figure 3). However, all supplier directions should be followed, including the use of a sedimentation cylinder with pure water that is used to store the device during standby. This procedure results in the precision of the size of the identified particle, specifically for clay materials.

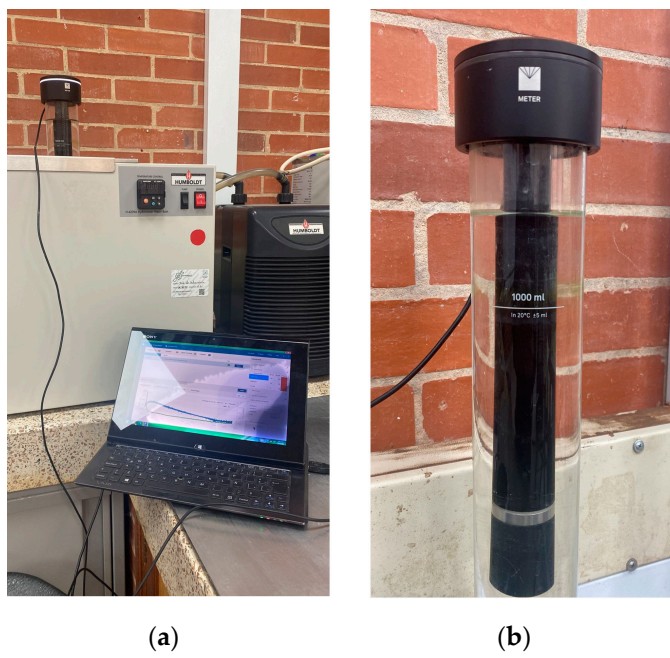

(**a**)　　　　　　　　(**b**)

**Figure 3.** Integral Suspension Pressure method (**a**) Measuring system and (**b**) ISP apparatus.

The pressure measured $p$ (Pa) at the maximum settling point, $L$ (m), at time $t$ (s) is obtained by integral calculus processes of the density as a function of the depth of the suspension. Since the density of fluid is constant with depth under isothermal conditions, the pressure time series at any precipitation depth are described by the Equation (3) [26]:

$$\rho(L,t) = \left(\rho_w + \left(m_{SHMP}/V_i\right)\right)gL + \rho_{add}g\int_0^L F(d)\,dz \tag{3}$$

where $g$ is the acceleration due to gravity, $\rho_w$ [kgm$^{-3}$] is the density of water, $m_{SHMP}$ [kg] is the mass of dispersant added, $V_i$ is the volume of the solution, and $\rho_{add}$ [kgm$^{-3}$] is the additional density due to the particular concentration. The corresponding mass fraction is given by the cumulative particle mass distribution function $F$, the integral calculation of which depends on the variation of $z$ from 0 to the maximum settling depth $L$.

For the determination of particle dimension exposed in Equation (4) [26], Stokes' law adequately describes the terminal precipitation velocity $u$ of settled spherical particles in laminar flow as a function of diameter $d$ (m) for any time $t$ at depth $z$ (Equation (4)) :

$$u(z,t) = \left(gt\left[\rho_s - \left(\rho_w + \left(m_{SHMP}/V_i\right)\right)\right]/18nz\right)d^2 \tag{4}$$

where $\rho_s$ [kgm$^{-3}$] is the density of the suspended particles, which Stokes indicated is assumed to be equal for each monogranular agent, $n$ is the viscosity of the fluid, and $V_i$ is the volume of the solution. The practical application of the ISP method is based on a much more precise obtaining of the PSD of the analyzed material, due to the high degree of repeatability between different samples of the same type of material. This technique will allow the clay soils to be adequately characterized, knowing the exact amount of clays and silts, especially in very fine soils such as montmorillonite. The hydrometer method can be deficient in this type of ultra-expansive material.

## 3. Results and Discussion

It is possible to objectively evaluate the application criteria by means of reproducibility, and this aspect is investigated in more detail below. Figure 4 illustrates a significant difference in reproducibility in the two measurement techniques for each of the three soil types. For technical reasons, the hydrometer allows the identification of a limited number of grain size classes. This is because the buoyancy nature of the element does not allow an adequate measurement of the finer particles that settle in the container. In the same way, the dependence on the human factor reduces the effectiveness of the method. As noted in Figure 4, the ISP method covers a much broader spectrum of particle sizes than the hydrometer due to the high sensitivity of its pressure sensor.

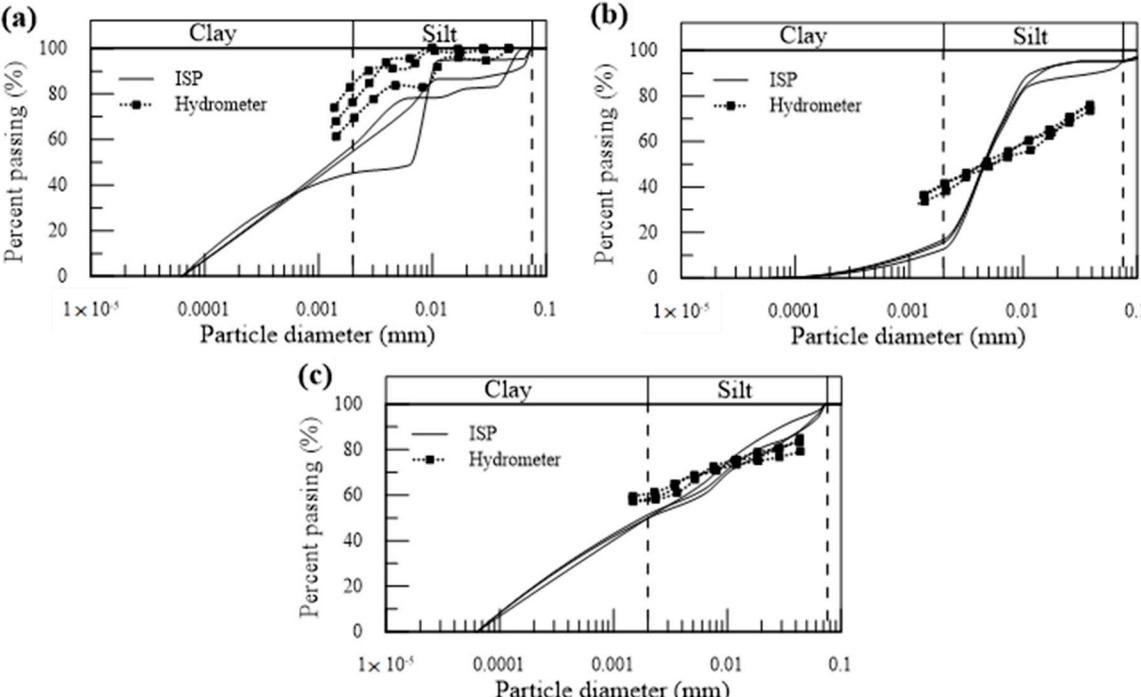

**Figure 4.** Particle Size Distribution (PSD) Curves Obtained by ISP and Hydrometer for (**a**) Bentonite, (**b**) Kaolin$_w$ and (**c**) Kaolin$_y$.

The ISP, in turn, performs a vertical sweep through the settling solution in the measuring cylinder. Therefore, the grain size data are derived from many pressure results, and a relatively high level of reproducibility can be expected. Additionally, the intrinsic ISP procedure is fully automated, which helps to reduce errors due to empirical execution.

The analytical range within which the diametric spectrum is measured varies considerably from one technique to the other. Both methods can measure particle sizes from approximately 1 to 75 μm, though the textural range is covered by ISP, with sweeps from 0.01 to 100 μm. On the other hand, a higher spectrum in particle size analysis with ISP is not necessary. Indeed, the application of each granulometric method is limited to a particular range of size classes. This implies dividing the sample into two granulometric fractions,

i.e., coarse, and fine-grained, with a boundary located at 75 μm. From this perspective, both methods implicitly consider the total mass of finest particles. Regarding the graphical outline of the PSD within the comparative scale, the descriptive trajectories reach significant differences for the hydrometer and PSI in bentonite and Kaolin$_w$ as well as high similitude in the Kaolin$_y$. It can be expected that the fine particle increment in these soils is partially responsible for the variability between curves, which is expected given that the effective hydrometer depth depends exclusively on the density of the suspension. Therefore, if the fine content within the soil is higher, the variations of concentration within the solution insubstantially influence the level of sedimentation recorded.

Although some materials are nominally referred to as clays (Kaolin$_w$ and Kaolin$_y$), several of them possess particle size distributions predominantly belonging to silt diameters. The slopes of the curves in the range of 2 to 0.4 μm are relatively smooth for the hydrometer and ISP. This indicates that the measurement in each sample is practically free of particles with intermediate diameters within this interval. For bentonite, the observations show considerable differences between the curves described by the hydrometer. This behavior indicates low levels of repeatability of this method for soils with clay contents above 80%. When specific portions of the ISP cumulative curve have steep PSD slopes at diameters above 2 μm, the hydrometer results for that same segment tends to reveal superior data to the relative percentages.

Through this study, it is possible to demonstrate the variability in the results of the particle size analysis according to the technique used. It is important to determine whether the variety in the mechanics of the data can directly affect the overall classification of the soil. Pedogenetics can be constructed according to several criteria but is generally appropriate according to the textural composition of the sediments. Figure 5 shows the three soils tested in this study and the position of each method in the texture triangle. This methodological proposal is based on the research carried out by [26,27].

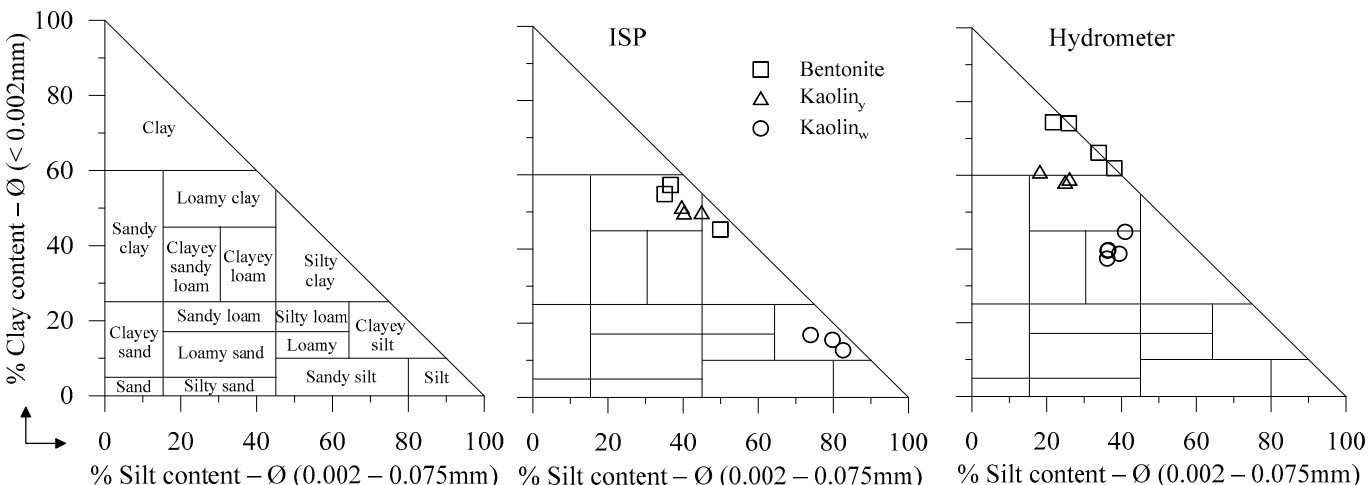

**Figure 5.** Texture triangle classes of measured soil samples obtained by ISP and hydrometer according to the USDA classification system.

Figure 5 shows that soils can be classified differently depending on the PSD determination technique [33]. The scenario is especially clear for Kaolin$_y$; here, the methods are placed in two different classes (clayey loam and clayey silt). This highly variable behavior is directly dependent on the high content of suspended colloids. In the case of the hydrometer, the measurements are generated by the changes in the liquid density derived from the sedimentation processes. This operating mechanism, mainly physical, is directly related to the size of the clay particles. When the proportion of colloids in suspension increases, the measurement capacity of the hydrometer decreases due to the slight density variation recorded. This limitation is more evident in soils with high content of particles with diameters less than 0.002 mm.

For the remaining soils, the fluctuation of results in each technique is generated within two approximately close sediment typologies. According to these results obtained, the reproducibility in the ISP method is relatively high in kaolinitic soils and tends to have a certain reduction in bentonite, at least for this investigation. In general, the method has high reproducibility due to the procedure carried out, which has little user participation, minimizing human error.

In general, the main source of variability can be derived from the more pronounced multimodality in the PSD curves obtained by ISP for the three soils compared to those obtained in their entirety by hydrometer. The maximum and minimum crossing points between PSD curves for each method were approximately common and well pronounced exclusively for soils with a predominant silt fraction (Kaolin$_y$), which can be explained by the discontinuities in the binding particles produced by the deficiency of the hydrometer when identifying the entire size spectrum. In other words, at the fine end, a particular density less than 1 μm in diameter does not influence the buoyancy of the hydrometer during the analysis, resulting in a higher value record of that particular value with respect to the ISP. Undoubtedly, the appropriate technique in the analysis influences the definition of PSD.

*Comparison of the PSD from the ISP and Hydrometer*

The correlations between the percentages of clay and silt obtained by the ISP and the hydrometer for the three soil typologies are presented in Figure 6. It was found that the ISP produced smaller clay fractions than the hydrometer for one soil sample (yellow kaolin). In contrast, the hydrometer produced a high proportion of clay for bentonite soil. Based on the equations listed in Table 5, the regression relationships for clay and silt were evaluated by the coefficient of determination ($R^2$).

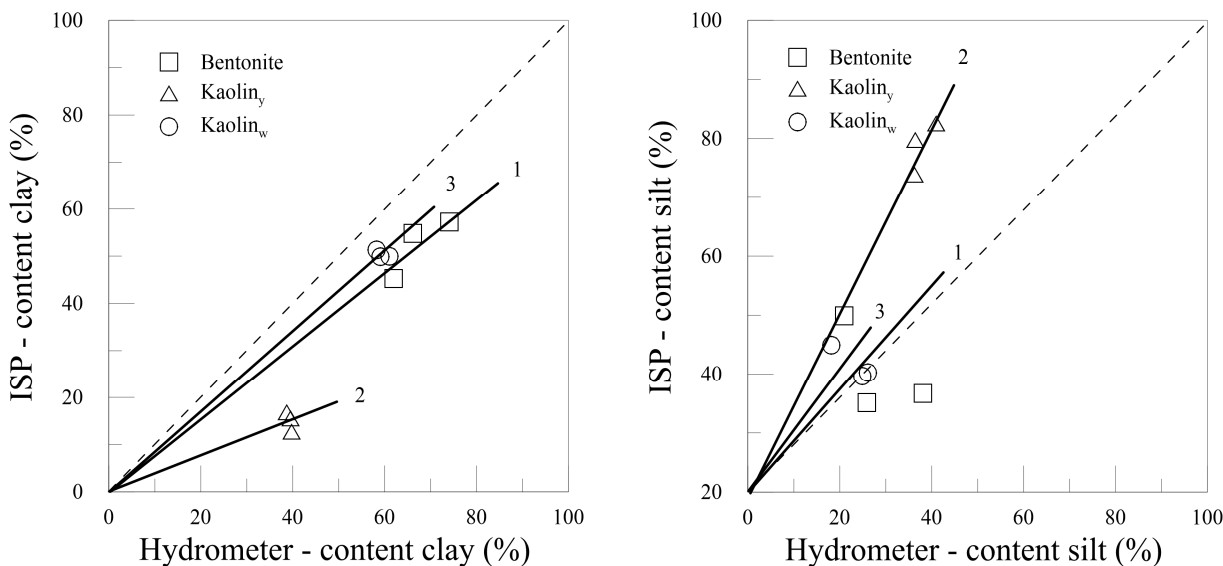

**Figure 6.** Scatter plots of the correlations of clay, and silt fractions obtained by hydrometry and ISP (Note: see Table 5 for the explanation of the numbers. The dotted line means a 1:1 slope).

The intercept term forced to zero of the results at the comparison points (clay and silt fraction) shows close agreement with ($R^2$) values above 80% in all correlation lines. This indicates that the ASTMD7928-21 and DIN 18,123 methods provide significantly similar granulometric results. The close agreement between the methods for the fractional typologies can be attributed to the fact that this statistic is determined under the same physical principle so that the influences generated by the compound errors in each method are minimal.

**Table 5.** Comparisons of the regression results for correlations of clay and silt fractions obtained by hydrometer and ISP.

| Number | Soil | Clay | | Silt | |
|:---:|:---:|:---:|:---:|:---:|:---:|
| | | Equation | $R^2$ | Equation | $R^2$ |
| 1 | Bentonite | y = 0.779x | 0.99 | y = 1.309x | 0.86 |
| 2 | $Kaolin_y$ | y = 0.382x | 0.98 | y = 2.076x | 0.99 |
| 3 | $Kaolin_w$ | y = 0.847x | 0.99 | y = 1.749x | 0.95 |

Each relation enables the obtention of the variation of the data with respect to both the trend line of each data set and the global trend line for y = x. The latter is used to analyze the trajectories of each set according to an assumed equal 1:1 state, which confirmed that samples with high silt contents may produce less variable PSD results in scenarios of superposition of both methods. On the other hand, the group analysis of all linear trends establishes a degree of proximity between trends equal to 80%. In these individual analytical observations, the ISP data could compare satisfactorily with the hydrometer data for a given size fraction. However, this formulation for a universal relationship between data from both methods requires further investigation. According to the analytical interpretation of the ($R^2$) results, a high degree of dispersion was observed in the measured clay contents when the ISP method was used. This behavior was analyzed by Nemes et al. (2020) [42] when comparing the results of the pipette with those obtained during the measurement of low clay contents with the ISP. However, for $Kaolin_y$, where the silt content is more significant than 40%, the ISP indicates a tendency to show higher values.

By directly comparing the data and making a merely pragmatic analysis within the scatter plots, the ISP methodology reveals lower values than those registered by the hydrometer for clay contents. In contrast, the proportion of silt appears lower in hydrometer measurements. In this type of graph, a match on a 1:1 line would infer that the two methods yield perfect results for two different extraction techniques, which is unlikely.

The performance of the techniques can also be studied in terms of the coefficient of variability between the individual results for each soil type. The coefficient of variation (CV), expressed as a percentage, is a widely used alternative classification quantity that assesses the reproducibility of a group of measurement methods or equipment. Assuming as a random variable the fractional components of clay and silt within each soil, which means a mean ($\mu$) greater than 0, and a standard deviation ($\sigma$) taken as the normal distribution between the data, the CV was estimated as the relationship between both statistical properties. An alternative way to rank the accuracy of the methods is by studying the grain size results in relation to the combination of all fractional components within each soil. Figure 7 shows the overall variation in the results between methods and each textural modulus within each soil sample, i.e., three replicate tests per soil to calculate CV. For soils with high silt content ($Kaolin_y$), the variability between techniques for fractional percentages is visibly limited. However, the greatest uncertainty is generated around silt contents, where in the soils studied, the coefficient of variability assumes data greater than 30% between methods, which indicated that the measurement variability may be related to the silty samples. This behavior may be justified since as hydrometer sensitivity decreases at the effective limit [43], some submicron particles may remain undetectable unless they contribute to aggregate formation in size above this limit, resulting in substantial limitation in silty content for this particular technique.

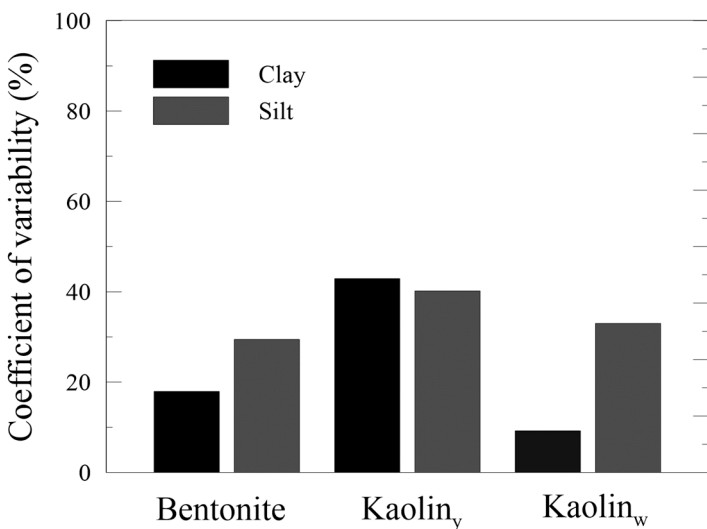

**Figure 7.** Coefficient of variability between soil as a function of the clay and silt contents.

### 4. Conclusions

Since the mechanism of the extended integral suspension pressure (ISP) method was used to evaluate the reproducibility of the particle size distribution (PSD) curve concerning the hydrometer derivative, the results of the current study indicated that samples with a strong influence of coarse particles or with a broad multimodal spectrum and PSD curves were more likely to produce high measurement variability. However, the general findings allowed the techniques to be classified under well-defined criteria. This classification is reflected through the physical principle on which these methodologies operate.

Although the final choice of a grain size technique depends on many factors, it can be concluded that the ISP mechanism generally scores better results for the different criteria evaluated in this study, which may establish it as a more optimized method in terms of PSD calculation for fine-grained soils [44]. The reasons for this analysis are the diametral range achieved, the low complexity in the operation protocol, and the freedom of grain orientation during the measurements. Additionally, the measurement speed is very high, and data processing is kept to a minimum. The method reproduces that the particles can settle at a typical speed that depends mainly on the fluid's viscosity and diameter. Therefore, the analysis procedure is considered valid when comparing two techniques based on the same principle, such as the hydrometer and the ISP.

Because the hypothesis aims to analyze the particle size distribution of the fine fraction of the samples, specimens with particles larger than 75 μm were not used. This emerging technique to obtain the PSD in soils (ISP) is advantageous due to its high precision, reproducibility, and repeatability features. This aspect starkly contrasts the hydrometer technique, which has shown deficiencies when evaluating highly plastic clays, such as montmorillonite, as discussed above. This traditional methodology records the data at different times and is taken by hand, even when the technician is not in the laboratory.

However, more extensive research is still required in different types of soils since divergences have been reported in the literature compared to already validated traditional techniques, for example, in detecting very fine sizes of clay particles. Therefore, there is still a need to analyze more studies with classical methods such as the hydrometer and the pipette, which allow establishing the ISP technique as a reliable alternative in obtaining the granulometric distribution in finely fractionated soils.

Due to its high performance and data accuracy, it is recommended to always use the ISP technique in soils with high silt or clay content.

**Author Contributions:** Conceptualization, M.C.O. and J.C.R.; methodology, M.C.O., J.C.R. and J.F.R.-R.; validation, M.C.O., J.C.R. and J.F.R.-R.; formal analysis, M.C.O., J.C.R. and J.F.R.-R.; investigation, M.C.O., J.C.R. and J.F.R.-R.; writing—original draft preparation, M.C.O.; writing—review and editing, M.C.O., J.C.R. and J.F.R.-R.; funding acquisition, J.C.R. All authors have read and agreed to the published version of the manuscript.

**Funding:** This research and the APC was funded by Universidad Catholica de Colombia under project 00491 of 2021.

**Institutional Review Board Statement:** Not applicable.

**Informed Consent Statement:** Not applicable.

**Data Availability Statement:** The data presented in this study are available on request from the corresponding author. The data are not publicly available due to due to restrictions of our institutions.

**Acknowledgments:** The authors are grateful for the support provided by the Universidad Católica de Colombia.

**Conflicts of Interest:** The authors declare no conflict of interest.

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
