# Peer review of "Synthesized Approach for Evaluating the Integral Suspension Pressure (ISP) Method and the Hydrometer in the Determination of Particle Size Distribution"

_sustainability, doi:10.3390/su15086847_

Round 1

Reviewer 1 Report

See my comments (mainly questions) in sticky notes in the pdf file of the manuscript attached.

Author Response

Dear,

Kittitat Leelarungroj

Editor.

Sustainability

First of all, we want to thank the reviewers and the Editor for such valuable observations, which undoubtedly resulted in the quality of the article. All comments, suggestions and corrections were taken into account and applied to this new version of the document.

Important:

Regarding the comments of the editor, these were corrected by responding to the suggestions of the reviewers.

Reviewer #1:

  • This title is not written in proper English: maybe: ... in the determination of particle size distribution

The title was rewritten as:

Synthesized approach for evaluating the Integral Suspension Pressure (ISP) method and the hydrometer in the determination of particle-size distribution

  • What is the novelty in this research that contribute to sustainability?

A paragraph is inserted in the abstract that tries to describe the novelty of the research.

Although the IPS method is considered promising and reliable in measuring the particle size distribution of the fine fraction, current research lacks comparisons with traditional methods. This aspect would help in the definitive validation of the technique and its use in practical engineering.

  • ??? (Questions in parentheses, it is assumed that the wording is not well understood)

The entire wording and style of the abstract is reviewed again.

  • If it is considered old, ISP was already justified to be better. Why to prove it again in a specific selection of the materials studied?

The reviewer's concern is better clarified in an earlier response. However, although the ISP is already considered better than other methods, investigations of comparisons with classical methods are still lacking.

  • It is a methodological part describing formerly known methods.

Yes, this part is considered important for non-expert readers. Although another reviewer suggested removing another methodological part related to the sedimentation theory.

  • What is the goal of the research? It is partly mentioned in the abstract, but should be here too.

The objective and novelty of this research was complemented in the abstract.

  • What is the sense of the table for the actual study?

The authors consider this table essential for the study, since it summarizes the references that present investigations comparing with traditional methods, as well as other comparisons between different techniques.

  • Why is this part necessary for the actual study?

This part of the document is removed at the suggestion of the current and another reviewer

  • The detailed description of these methods is not necessary for this study.

Although the section related to the classical theory of sedimentation was removed at the suggestion of the reviewers. The authors believe that the detailed explanation of the ISP is important, because it is not a method yet known by all readers.

  • What is the justification of this statement? Only the comparison with the other method?

This paragraph is inserted, replacing a previous sentence that did not express the analysis of figure 6.

By directly comparing the data and making a merely pragmatic analysis within the scatter plots, the ISP methodology reveals lower values than those registered by the hydrometer for clay contents. In contrast, the proportion of silt appears lower in hydrometer measurements. In this type of graph, a match on a 1:1 line would infer that the two methods yield perfect results for two different extraction techniques, which is unlikely.

  • What was the hypothesis of this study and was it accepted or not?

Basically, the hypothesis or research idea of this paper is based on the direct comparison of the ISP technique, which allows differentiating very small particle sizes; In relation to the hydrometer, which has deficiencies when measuring the particle size distribution in very fine soils such as montmorillonites. In this context the hypothesis was validated.

  • It is declared here that the results are specific and provide no sufficient general information

It was decided to modify this fragment of the conclusions paragraph, for a better understanding of the reader and the evaluator.

Because the study hypothesis is aimed at analyzing the particle size distribution of the fine fraction of the samples, specimens with particles larger than 75 um were not used.

Reviewer 2 Report

After went through the manuscript thoroughly following points were noted.

Ø  Clarify in the title whether it should be Integral Suspension method or Integral Suspension Pressure method

Ø  Lot of grammar mistakes throughout the manuscript have been noted. Please check the language thoroughly.

Ø  Line No. 64-75: Authors clearly stated in this paragraph (References 20-25) hydrometer is already proven as old method. ISP and hydrometer are compared in reference [1], which is a text book printed in 2001. There’s a recent publication where both ISP and hydrometer are compared. Wonder why Authors didn’t mention this publication in the table and also in the literature.

Acevedo, S. E., Contreras, C. P., Ávila, C. J., and Bonilla, C. A. (2021). Testing the integral suspension pressure method for soil particle size analysis across a range of soil organic matter contents. Int. Agrophys., 35(4), pp.357-363. https://doi.org/10.31545/intagr/144387

Authors must clearly state the advancement of the present work when compared to the above mentioned reference.

Ø  Line No. 100 – 101: “Today the existing studies that attempt to establish a relationship between hydrometer and ISP are meager”. This is very general statement. Try to do thorough literature review and mention the novelty of the present work clearly.

Ø   Line No. 109 – 140: It’s a basic theory about sedimentation. Need to be removed (section 1.1).

Ø  More than 10 references mentioned in the manuscript are before 1970. Is it really necessary to include all these references? Some of them are very basic theory about some engineering terms. For example, Ref. 54 is Theory about sedimentation, which is not required. Authors used too many references without adding proper information.

Ø  In R&D part, Figure 4 clearly shown that results of these 2 methods are not at all comparable, far away from each other. This again raise the same question, what’s the necessary to compare ISP and hydrometer methods?

Ø  Refer to Figure 6: The correlations are totally wrong. All the lines are mostly passing through only one point only. To do the correlation fitting the line must go through at least 3 points. These results need to be corrected properly.

Ø  Conclusion part is not very clear. For example: Since the methodology used in the technique is based on Stokes’ law - Incomplete sentence. Many sentences are not clear in the conclusion part.

Ø  Author’s contribution part is not written.

Ø  Many references are not necessary. Please carefully check and remove them.

Author Response

Dear,

Kittitat Leelarungroj

Editor.

Sustainability

First of all, we want to thank the reviewers and the Editor for such valuable observations, which undoubtedly resulted in the quality of the article. All comments, suggestions and corrections were taken into account and applied to this new version of the document.

Important:

Regarding the comments of the editor, these were corrected by responding to the suggestions of the reviewers.

Reviewer #2:

  • Clarify in the wheter it should be Integral Suspension Method or Integral Suspension Pressure Method

The title was rewritten as:

Synthesized approach for evaluating the Integral Suspension Pressure (ISP) method and the hydrometer in particle-size distribution

  • Lot of grammar mistakes throughout the manuscript have been noted. Please check the language thorough

Revised style and grammar again throughout the document

  • Line No. 64-75: Authors clearly stated in this paragraph (References 20-25) hydrometer is already proven as old method. ISP and hydrometer are compared in reference [1], which is a text book printed in 2001. There’s a recent publication where both ISP and hydrometer are compared. Wonder why Authors didn’t mention this publication in the table and also in the literature.

Acevedo, S. E., Contreras, C. P., Ávila, C. J., and Bonilla, C. A. (2021). Testing the integral suspension pressure method for soil particle size analysis across a range of soil organic matter contents. Int. Agrophys., 35(4), pp.357-363. https://doi.org/10.31545/intagr/144387

Authors must clearly state the advancement of the present work when compared to the above mentioned reference

This reference was analyzed in our investigation. In fact, it is one of our most important references. For some reason, due to a typing error, it was left out. However, it is the first reference in Table 1 and was confused with the first reference in the entire document [1] (text book printed in 2001). In our version of the paper (before MDPI), the presence of this reference is clearly observed. In the same way, a paper by the same authors from 2020 was also analyzed, with similar results (before [34], now [35]). The reference requested now is [34] and it appears first in Table 1.

[35] Contreras, C., Acevedo, S., Martínez, S., and Bonilla, C. (2020). “Evaluating the integral suspension pressure method for measuring the particle size distribution in soils with high organic matter content.” EGU General Assembly Conference Ab-stracts, EGU. on-line.

In accordance with the above, the inclusion of the reference and the respective analysis carried out in the paper is shown, which includes the paper suggested by the evaluator.

  • Line No. 100 – 101: “Today the existing studies that attempt to establish a relationship between hydrometer and ISP are meager”. This is very general statement. Try to do thorough literature review and mention the novelty of the present work clearly.

The following paragraph was inserted trying to respond to what was stated

Practically, the only studies that involve ISP and hydrometer in terms of direct comparison are those articulated in references [34,35]. Although it is a relatively recent methodology, the studies carried out show promising results. However, compared with a traditional technique, such as hydrometry, it reveals some discrepancies in the data, especially in clay-type fractions. The ISP has an advantage over the hydrometer, which is its high sensitivity to detect tiny particles due to its sensor and allows a pseudo-continuous curve. The hydrometer, as previously mentioned, presents deficiencies in very plastic clays, with the presence of montmorillonite. It is still necessary to carry out more investigations that allow comparing, with high reliability, the ISP with classical methods such as hydrometer and pipette.

  • Line No. 109 – 140: It’s a basic theory about sedimentation. Need to be removed (section 1.1).

Section 1.1 is removed, at the request of the evaluator. The references [55,56,57] also disappear.

  • More than 10 references mentioned in the manuscript are before 1970. Is it really necessary to include all these references? Some of them are very basic theory about some engineering terms. For example, Ref. 54 is Theory about sedimentation, which is not required. Authors used too many references without adding proper information.

Indeed, many of the references are from before 1970. However, this is because a table was prepared that summarizes the main works in the research field of the article in a table that helps the reader to understand the historical context of the problem.

  • In R&D part, Figure 4 clearly shown that results of these 2 methods are not at all comparable, far away from each other. This again raise the same question, what’s the necessary to compare ISP and hydrometer methods?

In previous studies [34,35], disharmony between both methods have already been reported. Mainly based on the technique how the data is obtained, although both are based on Stokes' law, as explained in the document. This means that in some data ranges there are notable differences in the results. However, we are talking about comparing a traditional technique, validated and expanded throughout the world, with a novel technique that can, theoretically, improve the output data in the particle size distribution. Due to the high precision of the pressure sensor and the almost continuous emission of data, which reduces human error (great disadvantage of hydrometry).

  • Refer to Figure 6: The correlations are totally wrong. All the lines are mostly passing through only one point only. To do the correlation fitting the line must go through at least 3 points. These results need to be corrected properly.

In the case of the regression analysis, a method of forcing to 0 was used, in which the tendency of the results to follow a 1:1 relationship was evaluated. The physical sense of this examination system is based on the assumption that for a zero silt or clay content in one method, the other must equally estimate a content equal to 0. To support the analysis carried out in this study, Kaszubkiewicz et al. (2020) perform the same type of regression for the same approach to determine the particle size curve.

  • Conclusion part is not very clear. For example: Since the methodology used in the technique is based on Stokes’ law - Incomplete sentence. Many sentences are not clear in the conclusion part.

A line of the conclusions is removed and a new paragraph is added to understand and shape the conclusion that is articulated with the objective of the investigation.

However, more extensive research is still required in different types of soils since differences with already validated traditional techniques have been reported in the literature, for example, at the detection level of clay particles. Therefore, there is still a need to be improved in comparisons with classical methods such as hydrometer and pipette, which allow establishing the ISP technique as a 100% reliable alternative in obtaining the particle size distribution in finely fractioned soils.

  • Author’s contribution part is not written.

Done

  • Many references are not necessary. Please carefully check and remove them.

 Done

Reviewer 3 Report

Topic of the paper is interesting and important as well as for the particle size distribution the non-direct methods are used. This means that only the comparative analysis between each of non-direct methods are possible, so any reasonaby planned investigations are valuable. Authors carried out a valuable analysis of two of the methods, but the result gives only a partial explanation of the observed differences. In my opinion more wide analysis should be carried out on  a larger number of soils of different particles and mineralogical composition, to exclude or prove some of the hypotesis.

I have found one important incoherence, which should be explained: on  Fig. 4, Kaoliny contains around 60% of clay fraction and on Fig. 5 the same Kaoliny contains around 40% of clay fraction. Please check carefully data! Discussion can be influenced by this mistake.

I have also some suggestions:

             I would not to point out the problem of not sufficient number of data from hydrometer analysis. Of course there are only single data available, but interpolation doesn’t influence the final results importantly and shouldn’t be stated as a real problem (see. Fig. 4).

             Because Bentonite is highly hydrophile mineral and increases its volume when moisturizing, did Authors take swelling effect during time of analysis it into account?

             I suggest not to use the word “overestimate” because it suggest that results one of the methods are more accurate than another (which means one of the methods in more objective in results, which is difficult to assume in this case).

I have found a few “technical” mistakes also:

38: it is worth to write that  75 mm is the separation diameter according to ASTM, because according to ISO this is 63 mm.

153: Fig. 2 should be fulfilled with the name of country and it should be stated what is the “scale 4 km”. Is it the length of white rectangular?

161: Table 2 – it should be “Gs” not “Sg” probably.

170: Table 3 – I do not understand the content of Table 3. Eg., is 89.17% of Bentonite in Kaolinw? Why there is only 6.23% Bentonite in Bentonite sample? Please make it more clear.

248: Vi – please use subscript (Vi).

Author Response

Dear,

Kittitat Leelarungroj

Editor.

Sustainability

First of all, we want to thank the reviewers and the Editor for such valuable observations, which undoubtedly resulted in the quality of the article. All comments, suggestions and corrections were taken into account and applied to this new version of the document.

Important:

Regarding the comments of the editor, these were corrected by responding to the suggestions of the reviewers.

Reviewer #3:

  • I have found one important incoherence, which should be explained: on Fig. 4, Kaoliny contains around 60% of clay fraction and on Fig. 5 the same Kaoliny contains around 40% of clay fraction. Please check carefully data! Discussion can be influenced by this mistake.

Figure 5 was corrected; the error was due to a confusion when drawing the graph. However, the values are correct.

  • I would not to point out the problem of not sufficient number of data from hydrometer analysis. Of course, there are only single data available, but interpolation doesn’t influence the final results importantly and shouldn’t be stated as a real problem (see. Fig. 4).

The following paragraph is inserted to avoid the discussion about the interpolation that can lead to confusion to the lectors

As noted in Figure 4, the ISP method covers a much broader spectrum of particle sizes than the hydrometer, due to the high sensitivity of its pressure sensor.

  • Because Bentonite is highly hydrophile mineral and increases its volume when moisturizing, did Authors take swelling effect during time of analysis it into account?

The following paragraph is inserted to consider the expansive effect of bentonite

In addition, in clays with a highly expansive character, as in this situation, the workability of the sample is complex. The specimen tends to clump together if the soil is not adequately hydrated. At this point, the advantage of the ISP over the hydrometer has already been mentioned. It is still necessary to carry out more investigations that allow comparing, with high reliability, the ISP with classical methods such as hydrometer and pipette.

  • I suggest not to use the word “overestimate” because it suggests that results one of the methods are more accurate than another (which means one of the methods in more objective in results, which is difficult to assume in this case)

The word "overestimation" was replaced by synonyms or expressions that explain the context related to the difference between the two methods.

  • 38: it is worth to write that 75 mm is the separation diameter according to ASTM, because according to ISO this is 63 mm.

Done.

  • 153: Fig. 2 should be fulfilled with the name of country and it should be stated what is the “scale 4 km”. Is it the length of white rectangular?

Indeed, many of the references are from before 1970. However, this is because a table was prepared that summarizes the main works in the research field of the article in a table that helps the reader to understand the historical context of the problem.

  • 161: Table 2 – it should be “Gs” not “Sg” probably.

Done

  • 170: Table 3 – I do not understand the content of Table 3. Eg., is 89.17% of Bentonite in Kaolinw? Why there is only 6.23% Bentonite in Bentonite sample? Please make it more clear.

The table was edited, since there was an error in the first column where the clay mineral that makes up each of the samples goes

  • 248: Vi – please use subscript (Vi)

Done.

Reviewer 4 Report

Dear Authors,

The article presents a technique for measuring particle size distribution in fine-grained materials.

The modern method used in this research bases the measurement on the change in pressure of the aqueous medium caused by the progressive settling of fine particles. Different materials were evaluated within the study to compare the results of the integral suspension pressure (ISP) method with a traditional approach widely used worldwide, such as hydrometers.

This emerging technique to obtain the PSD in soils known as ISP is advantageous due to its high precision, reproducibility, and repeatability features. This is especially compared to a hydrometer which is highly dependent on the user and human error (factors). Therefore, in my opinion, this is an important article and worth publishing.

 The article is written in good style. The experimental program and the results obtained were adequately described. The literature is comprehensive and quoted correctly. I am attaching minor errors (marked in yellow) that should be corrected.

I congratulate the Authors on good research.

Kind regards.

Author Response

Dear,

Kittitat Leelarungroj

Editor.

Sustainability

First of all, we want to thank the reviewers and the Editor for such valuable observations, which undoubtedly resulted in the quality of the article. All comments, suggestions and corrections were taken into account and applied to this new version of the document.

Important:

Regarding the comments of the editor, these were corrected by responding to the suggestions of the reviewers.

Reviewer #4:

The symbol for specific gravity was corrected.

The sieve No. 200 has a separation of 0.075 mm.

The subscripts in Table 3  were corrected.

The reference 15 was corrected

Round 2

Reviewer 1 Report

The authors significantly improved the manuscript taking all the suggestions of the reviewers into consideration.

Author Response

Thank you for your suggestions, they were very valuable to enrich the article.

Reviewer 2 Report

Some of the important comments mentioned in the previous review are not answered by the authors properly especially about the novelty/technical gap. I have following major concerns need to be answered.

Abstract line No. 18 - Although the IPS method is considered promising and reliable in measuring the particle size distribution…. When it is stated as promising and reliable method by the authors, why it should be compared with basic methods?

Abstract Line No. 20-21: Comparing the ISP and hydrometer measurements, coefficient of determination values higher than 80 and 90% were observed for silt, and clay, respectively – These results are totally unacceptable since not enough data points to calculate R2. The line passing through single point only in most cases (refer Fig.6).  The same point was raised about Fig.6 in previous review too.

Line No. 102: Today the existing studies that attempt to establish a relationship between hydrometer and ISP are meager – Again it’s a very general statement

Line No. 103-105: Practically, the only studies that involve ISP and hydrometer in-terms of direct comparison are those articulated in references [34,35]. Although it is a relatively recent methodology, the studies carried out show promising results – Authors concluded it’s a promising technique from the above studies. So where is the technical gap? The same question was raised in the previous review, but not answered by the authors properly.

Line No.442-443: It is especially compared to the hydrometer, which is highly dependent on the user – Sentence is not meaningful. Need to be corrected.

Line No.443-448: Both sentences are not clear.

How pipette method is mentioned here even though it is not considered in this study?

Line No. 450-451: is this sentence is correct? Seems to be some mistakes in this sentence

Line No. 451-454: Why authors insisting to have hydrometer technique in this sentence? What about ISP for this type of particles?

Author Response

First of all, we want to thank the reviewers and the Editor for such valuable observations, which undoubtedly resulted in the quality of the article. All comments, suggestions and corrections were taken into account and applied to this new version of the document.

Reviewer #2:

  • Abstract line No. 18 - Although the IPS method is considered promising and reliable in measuring the particle size distribution…. When it is stated as promising and reliable method by the authors, why it should be compared with basic methods?

The authors establish in the abstract that it is promising. Although it shows reliability in the results, it still deserves to be evaluated and validated against traditional methods expanded in the world, which were already validated a long time ago and widely used in consulting and teaching in universities. However, the entry of a novel method requires extensive studies to finish supporting its reliability.

  • Abstract Line No. 20-21: Comparing the ISP and hydrometer measurements, coefficient of determination values higher than 80 and 90% were observed for silt, and clay, respectively – These results are totally unacceptable since not enough data points to calculate R2. The line passing through single point only in most cases (refer Fig.6). The same point was raised about Fig.6 in previous review too.

Dear reviewer,

The authors consider your correction within the following limits:

  1. We understand that, although the number of data is limited to generate a global analysis of the behaviour of the results, according to Pal et al. (2019), to generate a regression analysis, the number of data should be n > 2.
  2. We consider the correction of the estimated reviewer, given that his clarification is correct, and we omit any global concept on regression analysis. However, given the rationale for limit 1, we believe it is important to present the regression results in a specific way.

  • Line No. 102: Today the existing studies that attempt to establish a relationship between hydrometer and ISP are meager – Again it’s a very general statement”

Due to the fact that there is some disagreement with this sentence that begins this paragraph, we believe that it is better to eliminate it and that the paragraph begins by mentioning that: “Practically, the only studies that involve ISP and hydrometer in terms of direct comparison-son are those articulated in references [34,35]…..”

  • Line No. 103-105: Practically, the only studies that involve ISP and hydrometer in-terms of direct comparison are those articulated in references [34,35]. Although it is a relatively recent methodology, the studies carried out show promising results – Authors concluded it’s a promising technique from the above studies. So where is the technical gap? The same question was raised in the previous review, but not answered by the authors properly.

The main difference between the results obtained here and those presented in the references [34, 35] is that the implementation of the ISP, in this case, was performed on samples with a high clay content. The granulometry of this type of soil is generally recorded with difficulty because the hydrometer fails to reproduce the data of small diameter particles since they do not generate significant differences in the density of the solution. Therefore, this study tests the ability of the ISP to define the mapping of soils with extreme physical characteristics.

  • Line No.442-443: It is especially compared to the hydrometer, which is highly dependent on the user – Sentence is not meaningful. Need to be corrected.

This sentence is changed for the following, because it can lead to mistakes on the part of the reader

 This aspect is in stark contrast to the hydrometer technique, which has shown deficiencies when evaluating highly plastic clays, such as montmorillonite, as discussed above. In this traditional methodology, the data is recorded at different times and taken by hand, even when the technician is not present in the laboratory.

  • Line No.443-448: Both sentences are not clear.

The writing is improved trying to express the idea in a better way

“However, more extensive research is still required in different types of soils since di-vergences have been reported in the literature compared to already validated traditional techniques, for example, in detecting very fine sizes of clay particles. Therefore, there is still a need to analyze more studies with classical methods such as the hydrometer and the pipette, which allow establishing the ISP technique as a 100% reliable alternative in obtaining the granulometric distribution in finely fractionated soils”

  • How pipette method is mentioned here even though it is not considered in this study?

The authors have stipulated to carry out a study later with different methods of obtaining the particle size distribution, not only with a pipette, but with other methods mentioned in the state of the art of this paper.

 Line No. 450-451: is this sentence is correct? Seems to be some mistakes in this sentence

The writing is improved trying to express the idea in a better way

However, more extensive research is still required in different types of soils since di-vergences have been reported in the literature compared to already validated traditional techniques, for example, in detecting very fine sizes of clay particles. Therefore, there is still a need to analyze more studies with classical methods such as the hydrometer and the pipette, which allow establishing the ISP technique as a 100% reliable alternative in obtaining the granulometric distribution in finely fractionated soils.

El evaluador tuien  

  • Line No. 451-454: Why authors insisting to have hydrometer technique in this sentence? What about ISP for this type of particles?

The evaluator is absolutely right to dispute this matter. Therefore, the authors decide to delete the last sentence, since it is not consistent with the rest of the paper. In addition, the wording of the last paragraph is improved for a better understanding of the final conclusion.

Round 3

Reviewer 2 Report

Appreciate the efforts of authors to improve the quality of the manuscript. The following points are noted in the revised version of the manuscript.

1.       Keyword – Integral suspension pressure method

2.       In table 1, summarize the methods which involve ISP only. Remove other methods.

3.       Line No. 386 - This behavior was analyzed by [61] when comparing the results of the pipette with those obtained during the measurement of low clay contents with the ISP. Must mention the authors name (Nemes et. al. ) before [61].

4.       Comparing the ISP and hydrometer measurements, coefficient of determination values higher than 80 and 90% were observed for silt, and clay, respectively. Authors removed these lines from abstract – But I am surprised that this particular point is not discussed anywhere in the results and discussion. Why?

5.       The line passing through single point / no points in most cases (refer Fig.6). The same point was raised about Fig.6 in previous review1 and review 2 also. The reason for stressing this particular point again and again is that major results and conclusion of this manuscript is based on the coefficient of determination values.

I partially agree with authors response to this point as mentioned by to Pal et al. (2019), to generate a regression analysis, the number of data should be n > 2.

But to draw any line used for analysis, it must pass through minimum 2 points. Where as this is not satisfied in the Fig. 6 (b), Lines 1 and 3 not at all passing through single point. The line can be changed in any direction which will completely change the R-square value.

Particularly this particular point must be addressed by the authors carefully.

6.       Conclusion can be written in more précised manner.

Author Response

Dear,

Kittitat Leelarungroj

Editor.

Sustainability

First of all, we want to thank the reviewers and the Editor for such valuable observations, which undoubtedly resulted in the quality of the article. All comments, suggestions and corrections were taken into account and applied to this new version of the document.

  1. Keyword – Integral suspension pressure method

Done.

  1. In table 1, summarize the methods which involve ISP only. Remove other methods.

Done.

  1. Line No. 386 - This behavior was analyzed by [61] when comparing the results of the pipette with those obtained during the measurement of low clay contents with the ISP. Must mention the authors name (Nemes et. al. ) before [61].

Done.

  1. Comparing the ISP and hydrometer measurements, coefficient of determination values higher than 80 and 90% were observed for silt, and clay, respectively. Authors removed these lines from abstract – But I am surprised that this particular point is not discussed anywhere in the results and discussion. Why?

Since that comment had been the focus of numerous comments, it was effectively removed from the manuscript. This decision was made based on the reviewer's constructive criticism, centered on the fact that the limited number of data in the linear regression could not generate such extensive analyses.

  1. The line passing through single point / no points in most cases (refer Fig.6). The same point was raised about Fig.6 in previous review1 and review 2 also. The reason for stressing The line passing through single point / no points in most cases (refer Fig.6). The same point was raised about Fig.6 in previous review1 and review 2 also. The reason for stressing this particular point again and again is that major results and conclusion of this manuscript is based on the coefficient of determination values.

I partially agree with authors response to this point as mentioned by to Pal et al. (2019), to generate a regression analysis, the number of data should be n > 2.

But to draw any line used for analysis, it must pass through minimum 2 points. Whereas this is not satisfied in the Fig. 6 (b), Lines 1 and 3 not at all passing through single point. The line can be changed in any direction which will completely change the R-square value.

Particularly this particular point must be addressed by the authors carefully.

In this case, the regression generates that type of R factor and fit because the analysis is forced to zero. Therefore, not all of the data is always taken when generating this type of linear regression that follows a 1:1 trajectory. In forced-zero (no intercept) regression, simple sums of squares are used, indicating that the slope estimator is more accurate, regardless of the data selection in the linear path. For this reason, the interpretations for the cases of the model with and without the R2-intercept are different. The use of regression without intercept must have a theoretical-physical foundation. In this case, it was selected since if one technique indicates the value of 0% (whether in clay or silt), the other technique must estimate a value close to 0, which validates the 1:1 regression theory. Also, since the 0-forced regression searches for the smallest standard error of the parameter estimate, the fit estimates a trajectory that does not always span the entire data set.

  1. Conclusion can be written in more précised manner.

A new revision of the writing and style of the conclusions was carried out